# Use of locum doctors in NHS trusts in England: analysis of routinely collected workforce data 2019–2021

Christos Grigoroglou ,[1] Kieran Walshe,[2] Evangelos Kontopantelis ,[3,4] Jane Ferguson,[2] Gemma Stringer,[2] Darren Ashcroft,[3,5] Thomas Allen [1,6]

For numbered affiliations see end of article.

**Correspondence to**
Dr Christos Grigoroglou;
christos.grigoroglou@
manchester.ac.uk

## ABSTRACT

**Objectives** Temporary doctors, known as locum doctors, play an important role in the delivery of care in the National Health Service (NHS); however, little is known about the extent of locum use in NHS trusts. This study aimed to quantify and describe locum use for all NHS trusts in England in 2019–2021.

**Setting** Descriptive analyses of data on locum shifts from all NHS trusts in England in 2019–2021. Weekly data were available for the number of shifts filled by agency and bank staff and the number of shifts requested by each trust. Negative binomial models were used to investigate the association between the proportion of medical staffing provided by locums and NHS trust characteristics.

**Results** In 2019, on average 4.4% of total medical staffing was provided by locums, but this varied substantially across trusts (25th–75th centile=2.2%–6.2%). Over time, on average two-thirds of locum shifts were filled by locum agencies and a third by trusts' staff banks. On average, 11.3% of shifts requested were left unfilled. In 2019–2021, the mean number of weekly shifts per trust increased by 19% (175.2–208.6) and the mean number of weekly unfilled shifts per trust increased by 54% (32.7 to 50.4). Trusts rated by the Care Quality Commission (CQC) as inadequate or requiring improvement (incidence rate ratio=1.495; 95% CI 1.191 to 1.877), and smaller trusts had a higher use of locums. Large variability was observed across regions for use of locums, proportion of shifts filled by locum agencies and unfilled shifts.

**Conclusions** There were large variations in the demand for and use of locum doctors in NHS trusts. Trusts with poor CQC ratings and smaller trusts appear to use locum doctors more intensively compared with other trust types. Unfilled shifts were at a 3-year high at the end of 2021 suggesting increased demand which may result from growing workforce shortages in NHS trusts.

## STRENGTHS AND LIMITATIONS OF THIS STUDY

⇒ National study covering every National Health Service (NHS) trust in England.
⇒ Data on locum doctors across NHS trusts have recently become available for the first time and this research study used this dataset.
⇒ Outcomes investigated included measures of locum use across trusts as well as outcomes pertaining to how trusts recruit locum doctors and how well trusts are able to fill their needs with locum doctors.
⇒ The data lacks some important information such as the types of medical specialties covered by locum doctors which would provide an improved understanding of the work that locums do.
⇒ Information on length of locum shifts would enable us to capture more accurately the level of locum intensity at the NHS trust level.

## INTRODUCTION

In the UK, challenges in the recruitment and retention of medical staff, including doctors of all grades, consultants, registrars and other doctors in training, in the National Health Service (NHS) have resulted in significant staff shortages.[1–4] In 2018, 43% of NHS consultant posts in general medicine which were advertised were not filled and 40% of consultants and 63% of higher specialist trainees said that rota gaps occurred on a daily or weekly basis,[5] while recent surveys of middle grade doctors reported that their workload has become unsustainable under current staffing levels.[4]

When faced with medical workforce shortages, NHS trusts can fill shifts using locum doctors recruited through locum agencies, third party organisations who contract healthcare professionals to work temporarily within the NHS, or through internal staff banks. Increases in NHS expenditure for agency staff led to NHS Improvement introducing a locum pay cap to curb agency expenditure and a weekly system for gathering data on locum usage by NHS trusts in 2016.[6] The new set of rules for agency staff resulted in a reduction in spending on locum agency doctors from £3.6 billion in 2015/2016 to £2.38 billion in 2019/2020,[7] although many hospital trusts have applied for extensions of these price caps to fill their workforce gaps.[4] Despite national information on expenditure, little is known about the extent of locum doctors working across NHS trusts,

in contrast to general practice where NHS Digital has published workforce data since 2015.[8 9] To date, no study has described the scale of locum usage in NHS trusts in England.

The aim of this study was to use NHS Improvement data to quantify and describe locum use, and its variation, for all acute, ambulance, community and mental health NHS trusts in England from January 2019 to December 2021. We describe the rate at which NHS trusts were able to fill locum shifts and whether NHS trusts find their locum workforce via their own NHS staff banks or via locum agencies. We explore regional variations for these measures and identify NHS trusts with the highest and lowest locum usage in 2019, as this was the year before the onset of the COVID-19 pandemic which was a period of substantial disruptions in the delivery of NHS services. Finally, we examine whether some NHS trust and population characteristics explain variability in locum use at the trust level in 2019.

## METHODS
### Patient and public involvement
This study conducted with the support of a patient and public involvement forum, with whom the design, analysis and results were discussed.

### Data
#### NHS trust temporary staff employment data
In England, NHS Improvement is responsible for setting out rules which trusts are expected to follow on temporary staff expenditure. The rules have a strong focus on providing support to trusts to reduce their expenditure and to move towards a sustainable model of temporary staffing. To fulfil this responsibility and support trusts, NHS Improvement collects information from all NHS trusts on their employment of temporary staff. These data are not published and were secured for research through a bespoke data-sharing agreement.

We analysed data on locum use for all NHS trusts in England between January 2019 and December 2021. Data record the weekly number of shifts that were filled by bank or agency locums for each acute, ambulance, community and mental health trust in England. A shift is defined as the period between the doctor commencing and finishing their work but the duration of shifts is not collected. Bank staff are defined as staff who are usually sourced in-house or from temporary staff banks such as NHS Professionals, which is the largest of these banks supplying temporary staff to NHS trusts.[10] Agency staff are defined as staff who are not on the payroll of the NHS organisation offering employment and are sourced from a third party agency.

NHS Improvement data record information on the number of shifts filled by temporary staff in all staff groups but we focus on the medical and dental staff group which includes the aggregate number of shifts, filled by temporary doctors working in NHS trusts. The data capture a snapshot of the weekly number of shifts done by doctors within hospital and community health services (HCHS) of the NHS, who are defined as all practising doctors who are registered with the General Medical Council — including general practitioners and dental staff—who are employed substantively by trust, that is, are on a trust's payroll. Information on the total number of doctor shifts that were filled by bank staff, the total number of shifts filled by agency staff and the total number of shifts requested by each trust in every reporting week, was provided. A detailed table of all the variables in the NHS Improvement data is provided in online supplemental material.

### NHS trust characteristics
We collected monthly data on all trusts' substantive employees represented as full-time equivalents (FTE) and trust annual job turnover data for the medical and dental staff group using the NHS Workforce Statistics database.[11] Trust type information and trust overall inspection ratings were obtained from the Care Quality Commission (CQC), which rates NHS trusts as outstanding, good, requiring improvement or inadequate.[12] Trust-level deprivation was derived using hospital admissions data from NHS Digital and aggregating inpatient postcode deprivation for each trust.[13] Trust-level vacancy rates were obtained in the form of advertised FTE roles for medical and dental staff, available from the NHS Vacancy Statistics from NHS Digital.[14] These trust characteristics were linked to temporary staffing data using unique trust identifiers and are discussed in detail in online supplemental material.

### Analyses
#### Outcome measures
##### Locum intensity
Our primary outcome measured locum intensity for each NHS trust in every reporting week. To calculate locum intensity, we combined bank and agency shifts to obtain the total number of shifts reported at trust level in every reporting week. We adjusted this weekly total by the size of the permanent medical and dental workforce in each trust, specifically, the total number of locum shifts was divided by permanent doctor FTE, (ie, FTE of NHS and community health hospital doctors, consultants, associate specialists, specialty doctors, specialty registrars, foundation doctors/postgraduate doctors) to give the locum intensity. The annual mean locum intensity was calculated over 12 months of data. A locum intensity of 0.25 indicates that the trust filled 0.25 locum shifts per week per FTE permanent doctor. We report locum intensity in this way because we do not know the length of the reported locum shifts and therefore cannot directly convert them into FTE. If we assume that one FTE permanent doctor typically works five shifts per week and that shift length for permanent doctors and locum doctors is broadly equivalent, then a locum intensity of 0.25 means that 5% of medical staffing in that week was provided by locums.

> **Box 1 Worked example of outcome measure calculations for Manchester University National Health Service (NHS) Foundation trust in 2019**
>
> To obtain the mean locum intensity for Manchester University NHS Foundation trust in 2019, we combined the number of bank and agency shifts to calculate the total number of filled shifts out of the number of shifts requested. For every reporting week in 2019, we divided the total number of shifts by the monthly permanent doctor full-time equivalent (FTE) reported the in that month. For example, in the week commencing 7 January 2019, Manchester University NHS Foundation trust reported 205 agency shifts and 283 bank staff shifts. We divided the total number of shifts (ie, 488) by the reported permanent doctor FTE in January (ie, 4378.8) to obtain a locum intensity value of 0.11, suggesting that for every one full-time doctor the trust had 0.11 locum doctor shifts that week. That would equate to 2.2% ((0.11/5)×100) of care provided by locums in that week if we assume five shifts per FTE.
>
> We calculated the proportion of shifts filled by agency staff by dividing the total number of agency shifts by the total number of all shifts (agency and bank) for each trust in every reporting week. For example, the proportion of agency shifts for Manchester University NHS Foundation trust in the week commencing 7 January 2019 was (205/488)×100=42%.
>
> We also had information on the number of shifts that each trust requested in every reporting week. For the same week, Manchester University NHS Foundation trust requested 574 bank and agency shifts but failed to fill 86 of these giving an unfilled rate of 15% ((86/574)×100).

We present a worked example of the algorithm that we used in the calculation of each outcome in box 1.

### Proportion of agency shifts

Our second outcome measured trusts' reliance on agency staff, which are more costly than bank staff. We divided the number of agency shifts by the total number of filled shifts for every trust in every reporting week. An annual mean proportion of agency shifts was then calculated for each trust over 12 months of data.

### Proportion of unfilled shifts

Our third outcome measures shifts that the trust was unable to fill. The total number of shifts requested by each trust in every week was provided by NHS Improvement. The number of filled shifts was subtracted from the number of shifts requested to obtain the number of unfilled shifts for each trust in each week. We calculated the proportion of unfilled shifts by dividing unfilled shifts by shifts requested. An annual mean proportion of unfilled shifts was calculated for each trust over 12 months of data. Trusts occasionally reported a higher number of shifts filled than requested. In these cases, we adjusted the number of shifts requested to reflect the number of total shifts filled in that week. These adjustments were made 811 times out of 11 450 (7.1%) trust-week observations in 2019.

Our analysis dataset contained information on locum intensity, proportion of agency shifts, proportion of unfilled shifts and trust characteristics for 229 acute, mental health, ambulance and community health trusts in 2019. Of these, three acute trusts and one ambulance trust did not report data on monthly doctor FTE, and one acute trust and seven ambulance trusts reported zero weekly locum returns in every reporting week. Eight ambulance, one acute, one mental health and one community trust reported zero agency shifts in every reporting week. We also explored variation in the three outcomes over time, with 224 and 221 trusts reporting bank and agency shift data to NHS Improvement, in 2020 and 2021, respectively.

Our first set of analyses was descriptive and we used ordered bar charts to show the distribution of locum intensity, proportion of agency shifts and proportion of unfilled shifts for all trusts in 2019–2021. Violin plots showed the geographical variation in each outcome across regions. We used spatial maps to illustrate the distribution of each outcome across all sustainability and transformation partnerships (STPs), local partnerships aiming to improve health and quality of care in the areas they serve. Analysis from 2019 to 2021 uncovered whether trusts reported changes over time in locum intensity, proportion of agency shifts and proportion of unfilled shifts, a period including a majority of the COVID-19 pandemic in England.

Our second set of analyses was inferential and employed three mean-dispersion negative binomial regressions to model locum intensity, proportion of agency shifts and proportion of unfilled shifts in 2019. Each model used robust standard errors with fixed-effects predictors for region (as categorical, to account for between region variation) and outcome-specific offset to model the rate of events for each outcome. Our dependent variables were: the mean number of locum shifts (offset: natural logarithm of mean permanent doctor FTE to model the rate of locum shifts per permanent doctor FTE); the mean number of agency shifts (offset: natural logarithm of the mean total shifts to model the rate at which a shift is filled by agency staff) and the mean number of unfilled shifts (offset: natural logarithm of mean shifts requested to model the rate at which a requested shift is left unfilled). Offsets are used as each dependent variable is derived from count data, where the value of the count is determined by the size of the workforce or exposure to locums. Our choice of negative binomial models over standard Poisson models was based on the presence of overdispersion in the three outcomes. We controlled for CQC inspection rating, trust type (NHS general acute trusts, NHS specialist trusts, mental health trusts and ambulance trusts), trust size (quintiles of trust permanent doctor FTE), turnover and vacancy rates and regional effects. Marginal effects were also calculated for the statistically significant coefficients, to estimate the absolute effects of those predictors on shift coverages.

The final dataset consisted of 197 trusts out of 229 trusts in 2019 with complete data for all covariates (8.6% of data were missing). We performed a sensitivity regression analysis excluding 25 ambulance and community trusts as these trusts tend to employ very small numbers of doctors relative to acute and mental health NHS trusts. The

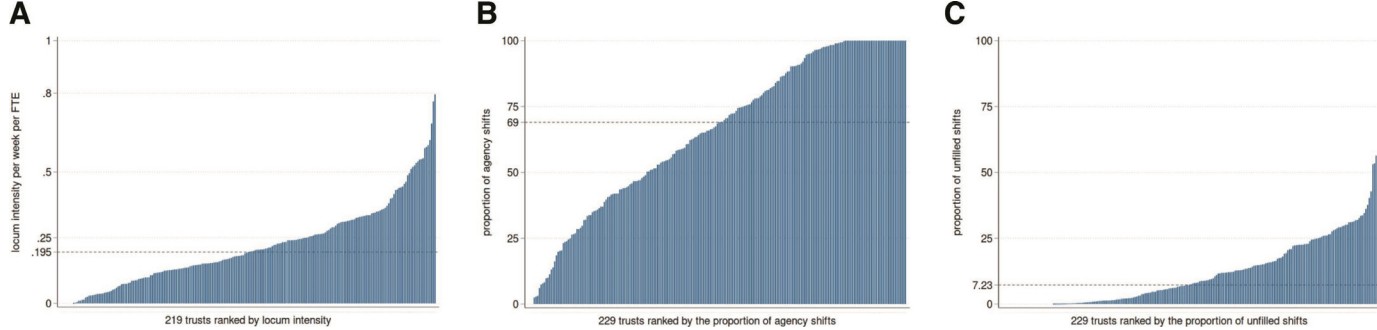

**Figure 1** Locum intensity, proportion of agency shifts and proportion of unfilled shifts in 2019, NHS trust level. (A) Mean locum intensity (bank and agency shifts combined) at trust level in 2019*†. (B) Mean proportion of agency shifts at trust level in 2019‡. (C) Mean proportion of unfilled shifts at trust level in 2019§. *Dash line indicates the median (0.195) locum intensity per week per FTE across 219 NHS trusts in 2019. †Ten trusts were excluded from the analyses due to very small or zero denominators (ie, low permanent doctor FTE). ‡Dash line indicates the median (69%) proportion of agency shifts per week across 229 NHS trusts in 2019. §Dash line indicates the median (7.23%) unfilled shift rate per week across 229 NHS trusts in 2019. FTE, full-time equivalent; NHS, National Health Service.

exclusion of ambulance and community trusts allowed us to examine the effects of deprivation, which could only be measured for acute and mental health NHS trusts. Stata V.16.1 was used for the principal data cleaning, management and analyses. We used the nbreg command with the exposure option.

## RESULTS
### Overall locum use
In 2019, total unadjusted locum use for all trusts in England was 2 004 485 shifts, of which 909 029 (45.3%) were bank shifts and 1 095 455 (54.7%) were agency shifts. Trusts requested 2 316 302 shifts with a trust mean of 208 per week (SD=258.3). The completeness of the data was good with 99% of all trusts reporting at least some locum use in any week.

### Locum intensity
Figure 1 plots the ranked mean locum intensity in 2019 for 219 NHS trusts in England showing significant variation in locum use across trusts. Mean locum intensity was 0.22 (SD=0.16) (median=0.195; 25th–75th centile=0.11–0.31) in 2019, indicating 0.22 locum shifts per permanent doctor FTE. Assuming five shifts per permanent doctor FTE, the average trust level locum intensity of 0.22 locum shifts per permanent doctor FTE was equivalent to 4.4% (ie, (0.22/5)×100) (25th–75th centile=2.2%–6.2%) of medical staffing provided by locums in 2019. Four ambulance trusts, three acute trusts and three community trust were not included in the descriptive analysis as they reported very low or zero permanent doctor FTE, and therefore, adjustments in their locum intensity could not be performed. The ranked rates of locum intensity in 2020 and 2021 are presented in online supplemental figure 1. We report the 10 trusts with the highest and lowest reported locum intensity in 2019 in online supplemental table 1.

### Proportion of agency shifts
The proportion of locum shifts filled by locum agency staff (rather than from staff banks) ranked from low to high at the trust level is depicted in figure 1B. The use of agency shifts varied substantially across trusts in 2019 with a mean of 66.1% (SD=28.5%; median=69%; 25th–75th centile=43.5%–95.8%). Half of trusts (109) reported 100% of shifts filled by agency staff at some point in 2019, of which 34 trusts reported that shifts were filled entirely by agency staff in every week. Eight ambulance, one acute, one mental health and one community trusts reported zero agency shifts in every reporting week in 2019. We present the ranked proportion of agency shifts for 2020 and 2021 in online supplemental figure 1.

### Proportion of unfilled shifts
In figure 1C, trusts are ranked from low to high on their proportion of unfilled shifts. Overall, trusts were able to fill the majority of their requested shifts either via bank or agency staff but we observed substantial variation. The mean proportion of unfilled shifts was 11.3% (SD=11.9%; Median=7.23%; 25th–75th centile=0.95%–18.1%). Seven ambulance and one acute trust did not request any shifts at any point in 2019. The ranked proportions of unfilled shifts for 2020 and 2021 are presented in online supplemental figure 1.

### Regional variation in locum use
In table 1, we present descriptive statistics on outcomes and trust characteristics at the regional level for 2019. Figure 2A–C shows regional variation in outcomes at the trust level in 2019. At the regional level, median locum intensity varied substantially from 0.13 (25th–75th centile: 0.08–0.2) in the South West of England to 0.26 (25th–75th centile: 0.15–0.35) in the Midlands (table 1 and figure 2A). We also observed large variation in the proportion of agency shifts across regions. Trusts in London filled the lowest proportion of shifts using agency staff with a median of 44.8%, (25th–75th

**Table 1** Descriptive statistics in 2019, by region*†

| | East of England | London | Midlands | North East and Yorkshire | North West | South East | South West |
|---|---|---|---|---|---|---|---|
| Locum intensity*, Median (25th–75th centile) | 0.19 (0.13 to 0.30) | 0.18 (0.08 to 0.28) | 0.26 (0.15 to 0.35) | 0.17 (0.08 to 0.35) | 0.21 (0.12 to 0.31) | 0.19 (0.11 to 0.27) | 0.13 (0.08 to 0.2) |
| Proportion of agency shifts (%), Median (25th–75th centile) | 78.1 (37 to 98.1) | 44.8 (26.6 to 87.5) | 75.6 (54.9 to 94.2) | 74.7 (51 to 99.5) | 65.3 (45.2 to 90.8) | 60 (34.3 to 88.9) | 77.9 (33.4 to 100) |
| Proportion of unfilled shifts† (% of requested shifts), Median (25th–75th centile) | 3.25 (0 to 13.1) | 11.6 (0.71 to 22.8) | 3.5 (0 to 16) | 3.9 (0 to 23) | 6.5 (0 to 21.6) | 4.8 (0 to 19.5) | 5 (0 to 17.6) |
| Full-time doctor FTE, Median (25th–75th centile) | 803.3 (385.4 to 1176.5) | 869.1 (398.9 to 2061.5) | 569 (220 to 1198) | 715.5 (268 to 1246.1) | 612.6 (321.9 to 1016.4) | 1013 (298 to 1317.7) | 701.1 (229 to 1082.2) |
| **Trust types** | | | | | | | |
| NHS general acute trusts (n) | 16 | 18 | 21 | 23 | 20 | 17 | 17 |
| Acute-NHS specialist trusts (n) | 1 | 5 | 3 | 1 | 6 | 1 | – |
| Mental health trusts (n) | 4 | 10 | 12 | 9 | 6 | 5 | 6 |
| Community health (n) | 3 | 2 | 4 | 1 | 2 | 5 | 1 |
| Ambulance service (n) | 1 | 1 | 2 | 2 | 1 | 2 | 1 |

Values indicated by bold represent the three outcomes under examination.
*Locum intensity is adjusted for mean total full-time doctor FTE in 2019.
†The proportion of unfilled shifts for trusts that reported a higher number of shifts filled than shifts requested was capped at 100%.
FTE, full-time equivalent; NHS, National Health Service.

centile=26.6%–87.5%) while trusts in the East of England filled the highest with a median of 78.1% (25th–75th centile=37%–98.1%) (table 1 and figure 2B). Trusts in the East of England filled requested shifts more successfully with unfilled shifts of 3.25% (25th–75th centile: 0%–13.1%) whereas trusts in London had unfilled shifts of 11.6% (25th–75th centile: 0.71%–22.8%) (table 1 and figure 2). Regional variation for the three outcomes in 2020 and 2021 is presented in online supplemental tables 2 and 3 and figure 2.

We investigated spatial variation within and between regions using spatial maps at the STP level (see online supplemental figures 3–5). Substantial variability was observed for all three outcome both within and between regions. High levels of locum intensity were concentrated in the Midlands, the North East & Yorkshire, and the East of England. The South East and South West ranked among the lowest in terms of locum intensity. High proportions of agency shifts were observed in areas in the Midlands, the East of England, and the South West. London had by far the lowest proportion of agency shifts. The proportion of unfilled shifts was high in areas in London, the Midlands and the North West and low in the East of England.

### Results from regression analyses

The regression analyses results using the three different outcomes are presented in table 2. The results are reported as incidence rate ratios (IRRs) for the coefficients of interest, followed by p values and standard errors in square brackets and 95% CIs in brackets. IRRs are defined as the number of exposed events (eg, number of locum shifts) divided by the number of unexposed events (offset—eg, permanent doctor FTE) in each time period and are essentially a ratio of two incidence rates. An IRR with a value greater than 1 indicates that the incident rate is higher in an exposed group compared with an unexposed group and the opposite is true for an IRR value less than 1. We focused on effect sizes rather than p values since statistical significance is more likely and can be less meaningful in large datasets such as the one we analysed.[15] Sensitivity analyses where we excluded ambulance and community trusts and examined the effects of deprivation on our three outcomes were nearly identical to the results from the main analyses. Deprivation did not appear to have any discernible effect on any of the three outcomes. The results from the sensitivity analyses are provided in online supplemental table 4 and the absolute differences in shift coverages for the statistically significant coefficients are provided in online supplemental table 5.

### Locum intensity

Results indicate that in 2019 trust size was a strong predictor of locum intensity. Using quintile 1 (ie, small trust size) as the reference group, our results showed significant reductions in locum intensity for medium and very large trusts with IRRs of 0.496 (95% CI 0.299 to 0.825) for quintile 3, and 0.347 (95% CI 0.187 to 0.644) for quintile 5. As an example of interpretation, comparing quintile 1

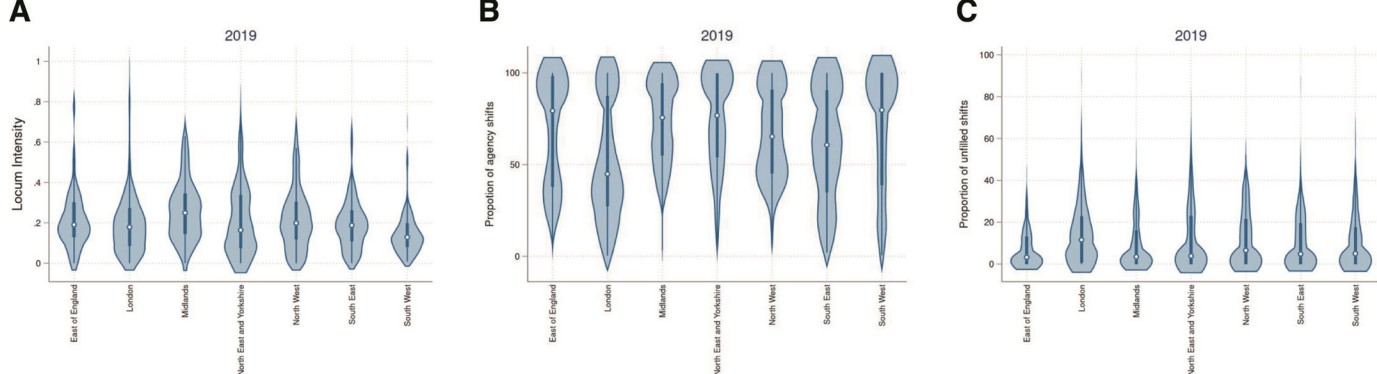

**Figure 2** Regional variation in locum intensity, proportion of agency shifts and proportion of unfilled shifts, 2019*. (A) Regional variation in locum intensity (bank+agency shifts combined) in 2019†‡. (B) Regional variation in the proportion of agency shifts in 2019§. (C) Regional variation in the proportion of unfilled shifts in 2019¶. *The thick blue line represents the interquartile range (25th–75th centile) and the thin line represents the rest of the distribution with upper/lower adjacent values. The white dot represents the median of the data. The distribution shape of the data is based on a kernel density estimation where wider sections of the plot represent a higher chance that members of the population of interest will take on a given value and where thinner section represent lower chance. †Figure includes data from 219 NHS trusts in 2019, adjusted for permanent doctor FTE. ‡Ten trusts were excluded from the analyses due to very small or zero denominators (ie, low permanent doctor FTE). §Figure includes data from 229 NHS trusts in 2019. ¶Figure includes data from 229 NHS trusts in 2019. FTE, full-time equivalent; NHS, National Health Service.

and quintile 3 suggests a locum intensity 50.4% lower for the medium size trusts. This was equivalent to 228.1 fewer weekly locum shifts for medium size trusts. NHS specialist trusts had a 71.5% lower locum intensity (IRR 0.285; 95% CI 0.174 to 0.468) than NHS general acute trusts and this effect was equivalent to 152.8 fewer weekly locum shifts for NHS specialist trusts. Ambulance service trusts had 55 times higher locum intensity than NHS general acute (IRR 55.43; 95% CI 20.56 to 149). However, this result is an artefact of the very low numbers of permanent doctors employed by ambulance trusts and the very small number of locum shifts filled when compared with other trust types. CQC ratings were strongly associated with higher locum intensity with trusts rated as inadequate or required improvement having 49.5% (IRR 1.495; 95% CI 1.191 to 1.877) higher mean locum intensity or 84.5 more weekly locum shifts than trusts rated good or outstanding. Staff turnover rates had negligible effects on locum intensity (IRR 1.015; 95% CI 1.009 to 1.021). Trusts in the South West had 40.25% lower locum intensity than trusts in London (IRR 0.575; 95% CI 0.361 to 0.915).

### Proportion of agency shifts
NHS specialist trusts and mental health trusts had 51% (IRR 1.510; 95% CI 1.086 to 2.100) and 57.6% (IRR 1.576; 95% CI 1.198 to 2.07) higher proportion of agency shifts than NHS general acute trusts. These effects were equivalent to 48.2 and 54.8 more weekly agency shifts for NHS specialist and mental health trusts, respectively. Ambulance service trusts had 96.7% lower proportion of agency shifts (IRR 0.033; 95% CI 0.008 to 0.147) than NHS general acute trusts. Trusts in the East of England had the highest proportion of agency shifts compared with trusts in London (IRR 1.525; 95% CI 1.167 to 1.993).

The effects of trust size on the proportion of agency shifts were not statistically significant.

### Proportion of unfilled shifts
NHS specialist trusts had 76.7% lower proportion of unfilled shifts when compared with NHS general acute trusts and this was equivalent to 23.77 fewer weekly unfilled shifts for NHS specialist trusts. Trusts in the East of England had 59.80 lower rates of unfilled shifts when compared with trusts in London (IRR 0.402; (95% CI 0.182 to 0.890)).

### Locum use during the COVID-19 pandemic
Figure 3 shows the mean agency, bank, unfilled and total shifts per week at the trust level in 2019 to 2021. Over time, the trust-level mean was 188.5 shifts per week (SD=205.8), of which 95.2 (SD=108.6) were agency shifts and 93.3 (SD=135.8) were bank staff shifts and the mean of unfilled shifts across all trusts was 38.5 (SD=85.2). Prepandemic, we observed small variability in the mean number of agency, bank and unfilled shifts. In March 2020, there was a steep decline (approximately 18%) in agency and bank shifts per trust as very few trusts reported locum use between March and April. In the third quarter of 2020, we observed an increase (approximately 15%) in agency and bank shifts per trust. In 2021, there was a steep steady increase in the mean number of unfilled shifts from 33.9 to 50.1 (47.8% increase) between May and June, which was sustained throughout 2021 and reached a peak of 69.2 unfilled shifts per trust in December 2021.

### DISCUSSION
### Summary
This study provides evidence on the extent of locum use and factors associated with locum use in NHS trusts in

**Table 2** Negative binomial regression analyses for the three outcomes in 2019, IRR*†

| Trust level aggregate FTE (reference group is quintile 1) | Locum intensity | Agency shifts | Unfilled shifts |
|---|---|---|---|
| | Reference group | Reference group | Reference group |
| Quintile 2 | 0.784 (0.527 to 1.676), <0.231 (0.159) | 0.945 (0.734 to 1.218), <0.662 (0.122) | 0.936 (0.449 to 1.952), <0.859 (0.351) |
| Quintile 3 | 0.496 (0.299 to 0.825)*, <0.007 (0.129) | 0.937 (0.675 to 1.301), <0.698 (0.157) | 1.848 (0.735 to 4.645), <0.192 (0.869) |
| Quintile 4 | 0.611 (0.349 to 1.072), <0.086 (0.175) | 0.883 (0.617 to 1.264), <0.497 (0.162) | 1.878 (0.704 to 5.011), <0.208 (0.940) |
| Quintile 5 | 0.347 (0.187 to 0.644)*, <0.001 (0.110) | 0.796 (0.530 to 1.195), <0.271(0.165) | 2.447 (0.826 to 7.251), <0.106 (1.356) |
| **Trust type (reference group is NHS general acute trust)** | **Reference group** | **Reference group** | **Reference group** |
| NHS specialist trust | 0.285 (0.174 to 0.468)*, <0.001 (0.072) | 1.510 (1.086 to 2.100)*, <0.014 (0.254) | 0.233 (0.091 to 0.598)*, <0.002 (0.112) |
| Mental health trust | 0.966 (0.628 to 1.486), <0.875 (0.212) | 1.576 (1.198 to 2.073)*, <0.001 (0.220) | 1.062 (0.508 to 2.221), <0.873 (0.400) |
| Ambulance service | 55.43 (20.56 to 149)*, <0.001 (27.96) | 0.033 (0.008 to 0.147)*, <0.001 (0.025) | 3.894 (0.453 to 33.14), <0.215 (4.272) |
| Community service | 1.443 (0.780 to 2.670), <0.243 (0.453) | 0.962 (0.641 to 1.445), <0.854 (0.199) | 1.360 (0.471 to 3.930), <0.570 (0.736) |
| **CQC ratings (reference group is trusts with good and outstanding services)** | **Reference group** | **Reference group** | **Reference group** |
| Inadequate and requiring improvement | 1.495 (1.191 to 1.877)*, <0.001 (0.173) | 1.044 (0.907 to 1.201), <0.550 (0.075) | 1.193 (0.789 to 1.804), <0.402 (0.251) |
| Trust level substantive doctor turnover rates | 1.015 (1.009 to 1.021)*, <0.001 (0.002) | 1.001 (0.997 to 1.003), <0.589 (0.001) | 0.995 (0.987 to 1.003), <0.248 (0.004) |
| Trust level vacancy rates (FTE) | 1.000 (0.999 to 1.001), <0.530 (0.005) | 0.999 (0.999 to 1.001), <0.948 (0.001) | 0.999 (0.997 to 1.001), <0.585 (0.001) |
| **Region (reference region is London)** | **Reference group** | **Reference group** | **Reference group** |
| South West | 0.575 (0.361 to 0.915)*, <0.019 (0.136) | 1.447 (1.098 to 1.907)*, <0.009 (0.204) | 0.687 (0.316 to 1.493), <0.343 (0.272) |
| South East | 0.701 (0.472 to 1.041), <0.078 (0.141) | 1.349 (1.047 to 1.736)*, <0.021 (0.175) | 0.524 (0.252 to 1.092), <0.085 (0.196) |
| Midlands | 1.041 (0.714 to 1.520), <0.832 (0.201) | 1.425 (1.126 to 1.804)*, <0.003 (0.172) | 0.548 (0.276 to 1.086), <0.085 (0.191) |
| East of England | 0.813 (0.533 to 1.240), <0.336 (0.175) | 1.525 (1.167 to 1.993)*, <0.002 (0.208) | 0.402 (0.182 to 0.890)*, <0.025 (0.163) |
| North West | 1.045 (0.705 to 1.550), <0.826 (0.210) | 1.327 (1.035 to 1.701)*, <0.026 (0.168) | 0.855 (0.412 to 1.773), <0.673 (0.318) |
| North East and Yorkshire | 0.754 (0.495 to 1.150), <0.191 (0.162) | 1.449 (1.120 to 1.875)*, <0.005 (0.191) | 0.575 (0.269 to 1.230), <0.154 (0.223) |
| Constant | 0.030 (0.152 to 0.601)*, <0.001 (0.105) | 0.436 (0.283 to 0.671)*, <0.001 (0.096) | 0.117 (0.038 to 0.357)*, <0.001 (0.066) |

*Model A included data on 220 trusts (observation) while models B and C included data on 214 trusts with robust standard errors.
†Coefficients can be interpreted as proportionate changes, for example, trusts in the North West had on average 4.5% higher locum intensity than trusts in London.
CQC, Care Quality Commission; FTE, full-time equivalent; IRR, incidence rate ratio; NHS, National Health Service.

England for the period 2019–2021. Our findings show that on average 4.4% of medical staffing in NHS trusts in 2019 was provided by locum medical staffing. Trusts with lower CQC ratings, acute trusts and smaller trusts had higher locum intensity. We observed moderate variability in locum use across regions and greater variability in the proportion of shifts filled by agency locums. During 2021, the proportion of shifts that were unfilled reached a 3-year high. Our findings can help inform NHS organisations about the extent of their locum use and provide for the first time important information about the drivers of locum use across NHS trusts. This can help with the effective planning of the NHS workforce by providing a better understanding of the make-up and spread of the locum medical workforce in England to aid recruitment in underperforming areas.

## Strengths and limitations of the study

The main strength of this study is the national scope and coverage of every NHS trust in England. For the first time,

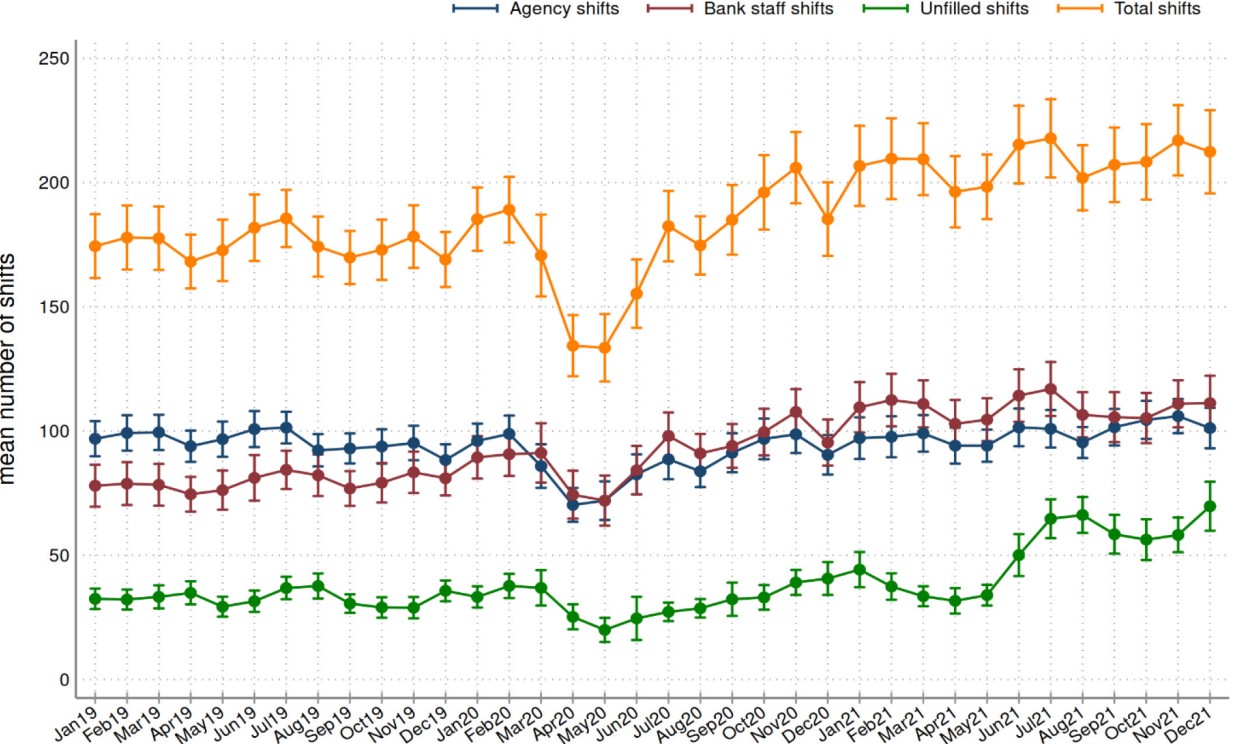

**Figure 3** Variation in mean number of locum shifts over time, 2019–2021*†. *Dots indicate the mean number of shifts across trusts for each month in the period 2019–2021 and the vertical lines represent the upper and lower 95% confidence limits. †The number of trusts that reported data in 2019, 2020 and 2021 were 229, 224 and 221, respectively.

using routinely collected data on locum use, we quantified the extent of locum use, sourced from agencies or banks, across all NHS trusts for the period 2019–2021. We also explored whether trusts were able to cover sufficiently for staff shortages and identified drivers of locum use at the trust level for the whole of England. We provide evidence on the extent of locum use across NHS trusts during the COVID-19 pandemic. Our analyses allowed us to control for measured trust and population characteristics.

However, this study has some important limitations which should be considered when interpreting the key findings. First, the NHS Improvement data do not reveal information on locum use by specialty and there may be substantial variations across specialties which we could not identify. Second, although NHS Improvement collects data on the number of locum shifts, it does not collect the shift duration, locum FTE or the number of shifts filled by permanent doctors which would allow a more straight-forward comparison with permanent doctor FTE. Therefore, we had to assume that shift lengths for permanent and locum doctors were broadly equivalent in order to estimate the proportion of medical staffing provided by locum doctors. Should data on the number of shifts filled by permanent doctors or data on locum FTE become available, this limitation could be addressed. Third, there may also exist variability in locum use between locums of different types (eg, infrequent or long-term locums) or durations apart from the agency/bank categories, which has been observed in general practice.[9] Some locums may be employed for several months[16] often to cover a vacancy

which has not been filled, while others may cover short-term absences such as illness for as little as one or two shifts and we did not have that information. Fourth, the dataset has no information on how well NHS trusts use their locum workforce such as the provision of adequate induction, training, supervision and feedback in accordance with NHS trusts in England guidance. Prior work[17] suggests that locum performance is driven more by organisational attributes such as these than by the characteristics of the locum doctors themselves. Fifth, the data do not contain any information on costs for locum doctors and we were therefore unable to estimate the extra financial costs of using agency locum to fill shifts.

### Interpretation of findings

The use of locum doctors is important because of the high level of spending it entails and because of concerns about the quality and safety of locum staffing arrangement.[18] Our study shows that the actual level of locum use, as a proportion of overall medical staffing, is relatively low on average, but varies considerably, with some trusts having much higher use of locums and some trusts relying overly on more expensive agency locums rather than using staff banks.

Some of this variation may be explained by organisational characteristics. For instance, larger trusts may be more able to cover workforce gaps within their own staff without needing locums, and specialist/tertiary trusts may find it easier to recruit and provide attractive work-places compared with general acute trusts. Mental health

trusts may face particular staffing shortages, which may explain the high level of agency locum use.

Our results show significantly higher locum intensity in trusts with worse CQC ratings (inadequate or requires improvement). It may be that these trusts find it harder to recruit and fill workforce gaps, but it could also be hypothesised that sustained high levels of locum use may impact quality and safety and hence affect CQC ratings.

The introduction of the first UK lockdown brought significant reductions in the numbers of both bank and agency locum doctors employed across NHS trusts, due to cancellations in elective care. However, shortly after, trusts started employing more locums likely in an effort to tackle excessive workload and increasing demand for healthcare services during the pandemic. Furthermore, in 2021, we observed an increase in the mean number of shifts filled by bank compared with the previous years and this was accompanied with a stable trend in agency shifts and an increase in the number of unfilled shifts. This suggests that trusts were meeting the increased demand with bank staff, which is in line with the new agency rules enacted by NHS Improvement in 2019 that aim to reduce reliance on agency staff.[19] Despite the increase in the mean number of total shifts, trusts appeared to be less able to fill the number of shifts they were requesting over the second half of 2021. This may suggest a persisting high workload for permanent doctors that trusts were unable to address with the use of locum doctors over that period.

**Author affiliations**
[1]Manchester Centre for Health Economics, Division of Population Health, Health Services Research and Primary Care, The University of Manchester, Manchester, UK
[2]Alliance Manchester Business School, Institute for Health Policy and Organisation, The University of Manchester, Manchester, UK
[3]NIHR School for Primary Care Research, Centre for Primary Care, Division of Population Health, Health Services Research, The University of Manchester, Manchester, UK
[4]Division of Informatics, Imaging and Data Sciences, The University of Manchester, Manchester, UK
[5]NIHR Greater Manchester Patient Safety Translational Research Centre, The University of Manchester, Manchester, UK
[6]Danish Centre for Health Economics, University of Southern Denmark, Odense, Denmark

**Acknowledgements** We would like to thank NHS Improvement and NHS Digital for the wealth of information they have collected and systematically organised, which made this study possible.

**Contributors** CG, TA and KW designed the study. CG extracted the data from all sources, performed the analyses and drafted the manuscript. KW, EK, JF, GS, DA and TA critically revised the manuscript. CG is the guarantor of this work and, as such, had full access to all the data in the study and takes responsibility for the integrity of the data and the accuracy of the data analysis.

**Funding** This study was funded by the National Institute for Health Research (NIHR) Health Service and Delivery Research programme (project reference number: NIHR128349).

**Disclaimer** The views expressed are those of the authors and not necessarily those of the NIHR or the Department of Health and Social Care.

**Competing interests** None declared.

**Patient and public involvement** Patients and/or the public were not involved in the design, or conduct, or reporting, or dissemination plans of this research.

**Patient consent for publication** Not applicable.

**Provenance and peer review** Not commissioned; externally peer reviewed.

**Data availability statement** Data may be obtained from a third party and are not publicly available. Data can be obtained from NHS Improvement under a special license and are not freely available.

**ORCID iDs**
Christos Grigoroglou http://orcid.org/0000-0003-1621-8648
Evangelos Kontopantelis http://orcid.org/0000-0001-6450-5815
Thomas Allen http://orcid.org/0000-0002-2972-7911

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
