## [Reviewer comments · BMJ Open]

ARTICLE DETAILS

TITLE (PROVISIONAL)	The use of locum doctors in NHS trusts in England: analysis of routinely collected workforce data 2019 – 2021.
AUTHORS	Grigoroglou, Christos; Walshe, K; Kontopantelis, Evangelos; Ferguson, Jane; Stringer, Gemma; Ashcroft, Darren; Allen, Thomas

VERSION 1 – REVIEW

REVIEWER	Weeks, WB Dartmouth College
REVIEW RETURNED	12-Jul-2022

GENERAL COMMENTS	This is an interesting and well-written paper that examines, for the first time, the degree to which locums doctors in NHS trusts are used. The methods are clear, the results are clearly presented, and the discussion makes sense. Limitations are articulated. I have a few suggestions that may help improve clarity further. It's a bit difficult to understand the logic of the several measures without first seeing the box. The authors might introduce the outcome measures section with the statement "a worked example of the algorithm....", then go into the description of the measures. Within the box, there is a typo (page 17, line 11). I believe it should be 'doctor FTE reported in that month.' I'm having trouble reconciling the statement in the abstract that 'over time, 2/3 of locum shifts...' with the results section, page 8, line 4 '45.3% were bank shifts and 54.7% were agency.' I see lower in that page (line 36) the 2/3 reference. Unless I'm misreading, those should all be consistent: they all represent the proportion of locum use that is agency vs. bank. Perhaps the distinction is in 'total UNADJUSTED' on line 3, page 8. If so, the authors should clarify in the abstract and the other sections that the 2/3 is an adjusted number. Page 7, line 44 in the parentheses, I believe the authors mean that '8.6% of data were missing.' While the authors report IRRs that have very large percentage differences, it would be helpful to also provide an estimate of the
---

	absolute differences in shift coverages, which may not be quite as dramatic.
REVIEWER	Ejebu, Ourega-Zoé University of Southampton, School of Health Sciences
REVIEW RETURNED	14-Nov-2022
GENERAL COMMENTS	Thanks to the authors for giving me the opportunity to review their paper. It was very interesting and I hope my comments will be useful. BW

VERSION 1 – AUTHOR RESPONSE

Reviewer(s)' Comments to Author:

Reviewer: 1

The papers uses NHS workforce routinely collected data to explore the use of locum doctors in NHS trusts in England. It is a very interesting paper that uses objective data to quantify locum uses across NHS trust in England.

Abstract

Objectives: The authors should briefly define what locum doctors are and later in the main manuscript how they are different from other types of doctors

Response: We have amended the abstract objectives to briefly define locum doctors. The new statement in the objectives reads “Temporary doctors, known as locum doctors play an important role in the delivery of care in the NHS”.

Conclusion: there is a typo. It should read “appear” instead of “app1ear”

Response: We have corrected the typo.

Introduction

Page 5 - Lines 6 to 13: The authors use various words to describe medical staff which makes it difficult to understand who they are referring to. For instance, in the same paragraph, you describe medical staff, NHS consultants, higher specialist trainees and middle-grade doctors. The author should perhaps briefly explain that the medical staff is diverse to avoid confusion.

Response: Thank you, this is valid point. We have added a statement in p.5 par1. to clarify this. The statement reads, “In the UK, challenges in the recruitment and retention of medical staff, including doctors of all grades, consultants, registrars and other doctors in training,”

Page 5 - Line 24: It may be too early, but would the authors have the figure for the year 2019/20 or 2020/2021 (i.e. during the pandemic) to realise the extent to which (if any) Covid impacted expenditures on locum doctors

Response: We have added the figure about the reduction of spending in 2019/20 which was very similar to the figure in 2018/19.

Page 5 - Line 41: why not in 2021 too?

Response: We selected 2019 as the primary year of analyses for several reasons. First, following the implementation of lockdown measures during the first wave of the COVID-19 pandemic, services in the NHS were disrupted with dramatic reductions in healthcare utilisation, hospital admissions and

diagnostic and imaging procedures, (1, 2) and with many UK hospitals focusing solely on managing COVID-19 patients. During this period, locum working was severely affected, where redeployment of NHS permanent staff, on-hold services, cancellations of annual leave and the lack of demand at the various COVID-19 hospitals resulted in less demand for locum doctors, for some of whom, work disappeared overnight. (3) This lack of demand for locum doctors during the first UK lockdown is evident in figure 3 where we show the variation in the mean number of locum shifts in 2019-2021, but has also been documented in our qualitative work which is underway. Second, in December 2021 when the UK was recording the highest COVID-19 case rates since the beginning of the pandemic, the demand for locum doctors in NHS trusts peaked, as shown again in figure, perhaps due to rising waiting times and increased workload for all NHS doctors. We therefore observed patterns of instability in the delivery of locum doctor services across NHS trusts in 2020-2021, and made the decision to analyse locum data for that period, descriptively (additional descriptive analysis is contained in the supplementary material). Third, some of the drivers of locum use that were used in our negative binomial models, such as CQC ratings, trust size and trust type may not have been relevant during the pandemic and interpretation would have been difficult. We have added a statement to clarify the use of 2019 data. In p.5 par.3 the statement now reads “We explore regional variations for these measures and identify NHS trusts with the highest and lowest locum usage in 2019, as this was the year before the onset of the COVID-19 pandemic which was a period of substantial disruptions in the delivery of NHS services.

Page 6 lines 23-29: The authors should be clearer as to which data and analyses are presented in their paper. It is only at this stage that you are referring to “dentists” as it was not mentioned earlier in the manuscript. What are the rationales for presenting data for doctors and dentists altogether? To my knowledge dentists are not doctors in the UK, so it’s not clear why results should be presented together. Finally, the literature and aims should be re-defined on the basis that your analysis includes data on dentists.

Response: This is a very good point, thank you. In the UK, the majority of NHS staff – approximately 1.2 million full-time equivalents – work in ‘hospital and community services’ (HCHS) as direct employees of NHS trusts providing acute, ambulance, mental health and community services. (4) This group also includes 150,000 staff who are substantive trust employees (i.e. they are on the trust’s payroll) and work in general practice, community pharmacies and dentistry.(5) Therefore, the NHS Improvement database provides data for all trusts’ substantive employees in every staff group, including our group of interest, defined as the medical and dental staff group. High street dentists are not included in the data. We understand that this was confusing, and we have amended the text to reflect this. In p.6 par. 3 the statement now reads, “The data capture a snapshot of the weekly number of shifts done by doctors within hospital and community services (HCHS) of the NHS, who are defined as all practising doctors who are registered with the General Medical Council (GMC) – including some GPs and dental staff – who are employed substantively by trusts i.e. are on a trust’s payroll”. We would also like to highlight that dental staff, concerns only a very small fraction of the NHS trust workforce. We have not changed the aims as we think this is likely to mislead the reader given the dental group is very small and not our intended focus – we hope the additional text helps clarify this.

Proportion of unfilled shifts Page 8 line 24: Could the authors explain why they’re only using the year 2019 to run the negative binomial model on locum intensity? Especially if you’re interested in the “between region variation”. Using the years 2020 and 2021 may shed more light on such “between variation”. Using a mixed effect negative binomial may also be of interest by having the region variable as the “random effect” if you suspect some clustering at the region level. The author should also briefly explain the rationales for using the offset. They are simply quoted without proving a justification for their choice.

Response: Thank you for your suggestions. For the reasons outlined in our response to your fifth comment, we ran the negative binomial model for all outcomes, using only the year 2019. Of course, if the reviewer would like to see the results from a model using all years, we can add them in the next

revision. However, we feel that the instability that was observed during 2020 and 2021 across NHS services would make the interpretation of results problematic.

We have used a mixed effect negative binomial model with region as a random effect and the results from the mixed effects and primary negative binomial model are provided below. Please note, the model below (Table 1) uses the total FTE of all clinically qualified staff who are substantive employees of NHS trusts following your next suggestion to use this as a measure for trust size in the place of permanent doctor FTE. We can supply results from the model using permanent doctor FTE as a measure of trust size, if needed. The results from the two models were almost identical, however, the likelihood ratio test (LR) favoured the negative binomial model over the mixed effects model with a region specific random (LR test for mixed effects model vs negative binomial model = 0.68 [Pr>= chibar2 =0.2050]); indicating that there was no significant variation in the number of locum shifts between regions. We therefore decided to keep the negative binomial model as the primary model in our analyses. Similar results were observed for the number of agency shifts (LR test for mixed effects model vs negative binomial model = 0.90 [Pr>= chibar2 =0.1717]) and the number of unfilled shifts (LR test for mixed effects model vs negative binomial model = 0.78 [Pr>= chibar2 =0.1937]).

Regarding the choice of offsets, we added a statement to justify their use in the text. In p.9 par.3 the statement reads “Our dependent variables were: the mean number of locum shifts (offset: natural logarithm of mean permanent doctor FTE to model the rate of locum shifts per permanent doctor FTE); the mean number of agency shifts (offset: natural logarithm of the mean total shifts to model the rate at which a shift is filled by agency staff); and the mean number of unfilled shifts (offset: natural logarithm of mean shifts requested to model the rate at which a requested shift is left unfilled). Offsets are used as each dependent variable is derived from count data, where the value of the count is determined by the size of the workforce or exposure to locums.”

Table 1: Mixed effects regression analyses for the three outcomes in 2019, IRR a,b

Locum Intensity

Agency shifts Unfilled shifts

Trust level aggregate FTE (all qualified clinical staff)

(reference group is quintile 1) Reference group Reference group Reference group

Quintile 2 0.712 (0.497 to 1.020), <0.065 [0.131] 0.921 (0.734 to 1.155), <0.480 [0.107] 0.946 (0.504 to 1.776), <0.863 [0.304]

Quintile 3 0.731 (0.508 to 1.054), <0.093 [0.136] 0.940 (0.746 to 1.183), <0.599 [0.110] 1.312 (0.684 to 2.516), <0.414 [0.435]

Quintile 4 0.736 (0.507 to 1.071), <0.109 [0.140] 0.941 (0.741 to 1.193), <0.615 [0.114] 1.265 (0.652 to 2.454), <0.487 [0.427]

Quintile 5 0.446 (0.287 to 0.694)*, <0.001 [0.100] 0.850 (0.635 to 1.137), <0.275 [0.126] 1.354 (0.615 to 2.983), <0.452 [0.545]

Trust type

(reference group is NHS general acute trust) Reference group Reference group Reference group

NHS specialist trust 0.347 (0.216 to 0.560) *, <0.001 [0.084] 1.395 (1.012 to 1.923), <0.042 [0.228] 0.260 (0.108 to 0.621)*, <0.002 [0.115]

Mental health trust 1.336 (0.991 to 1.801), <0.057 [0.204] 1.577 (1.307 to 1.903)*, <0.001 [0.151] 0.778 (0.468 to 1.296), <0.336 [0.202]

Ambulance service 73.43 (29 to 186.1) *, <0.001 [34.84] 0.035 (0.008 to 0.153)*, <0.001 [0.027] 2.482 (0.328 to 18.78), <0.378 [2.563]

Community service 2.266 (1.421 to 3.612)*, <0.001 [0.540] 0.974 (0.713 to 1.329), <0.869 [0.154] 0.843 (0.370 to 1.926), <0.687 [0.355]

CQC ratings

(reference group is trusts that provide good and outstanding services) Reference group Reference group Reference group

Inadequate and requiring improvement 1.455 (1.155 to 1.831)*, <0.001 [0.171] 1.033 (0.898 to 1.188), <0.646 [0.074] 1.021 (0.686 to 1.519), <0.918 [0.206]

Trust level substantive doctor turnover rates 1.015 (1.009 to 1.020)*, <0.001 [0.003] 1.001 (0.998 to 1.004), <0.580 [0.001] 0.996 (0.988 to 1.004), <0.397 [0.004]

Trust level vacancy rates [full-time equivalent (FTE)] 1.000 (0.999 to 1.001), <0.530 [0.005] 0.999 (0.999 to 1.001), <0.350 [0.001] 1.001 (0.998 to 1.002), <0.645 [0.001]

constant 0.192 (0.131 to 0.284)*, <0.001 [0.037] 0.610 (0.474 to 0.785)*, <0.001 [0.078] 0.103 (0.055 to 0.764)*, <0.001 [0.033]

a Model A included data on 220 trusts (observation) while models B and C included data on 214 trusts with robust standard errors.

b Coefficients can be interpreted as proportionate changes, for example, trusts that were rated as inadequate and requiring improvement had 45.5% higher locum intensity than trusts that were rated as having good and outstanding services.

Page 8 line 38 I wonder if only using trust permanent doctor FTE is enough to “measure” the trust size. Would it be worth considering the number of permanent nursing staff as well, as they represent the largest health workforce group?

Response: Thank you for your suggestion. We had initially tested our models for the inclusion of quintiles for all qualified clinical staff, including all HCHS doctors, nurses and health visitors, midwives, ambulance staff, scientific therapeutic and technical staff to measure trust size. We include the results from this analysis in our responses (Table 2). However, we chose not to provide these in the original submission because the model with quintiles of doctor FTE to measure trust size, performed slightly better according to the Akaike (AIC) and Bayesian (BIC) information criteria which are typically used to inform model selection in these cases. The negative binomial with doctor FTE as a measure of trust size minimised the AIC=2,584.11 and BIC=2,648.59 compared to the negative binomial model with all clinically qualified staff FTE as a measure of trust size, AIC=2,585.46 and BIC=2,649.93.

Table 2: Negative binomial regression analyses for the three outcomes in 2019, IRR a,b

Locum Intensity

Agency shifts Unfilled shifts

Trust level aggregate FTE (all qualified clinical staff)

(reference group is quintile 1) Reference group Reference group Reference group

Quintile 2 0.716 (0.502 to 1.022), <0.066 [0.130] 0.924 (0.740 to 1.155), <0.490 [0.105] 1.084 (0.568 to 2.068), <0.806 [0.357]

Quintile 3 0.762 (0.533 to 1.090), <0.137 [0.139] 0.938 (0.748 to 1.176), <0.578 [0.108] 1.039 (0.725 to 2.654), <0.322 [0.459]

Quintile 4 0.731 (0.506 to 1.057), <0.096 [0.137] 0.946 (0.748 to 1.197), <0.646 [0.114] 1.425 (0.724 to 2.801), <0.305 [0.491]

Quintile 5 0.439 (0.284 to 0.682)*, <0.001 [0.098] 0.845 (0.633 to 1.128), <0.254 [0.125] 1.558 (0.701 to 3.462), <0.277 [0.634]

Trust type

(reference group is NHS general acute trust) Reference group Reference group Reference group

NHS specialist trust 0.311 (0.194 to 0.500)*, <0.001 [0.075] 1.514 (1.109 to 2.066)*, <0.009 [0.228] 0.194 (0.076 to 0.499)*, <0.002 [0.093]

Mental health trust 1.316 (0.975 to 1.779), <0.073 [0.202] 1.661 (1.386 to 1.990)*, <0.001 [0.153] 0.671 (0.399 to 1.128), <0.132 [0.177]

Ambulance service 79.81 (31.66 to 201.3)*, <0.001 [37.66] 0.037 (0.008 to 0.157)*, <0.001 [0.027] 2.451 (0.328 to 18.3), <0.382 [2.514]

Community service 2.093 (1.306 to 3.353)*, <0.001 [0.503] 1.025 (0.751 to 1.398), <0.877 [0.162] 0.936 (0.398 to 2.198), <0.879 [0.408]

CQC ratings

(reference group is trusts that provide good and outstanding services) Reference group Reference group Reference group

Inadequate and requiring improvement 1.450 (1.155 to 1.820)*, <0.001 [0.168] 1.030 (0.897 to 1.183), <0.674 [0.073] 1.087 (0.732 to 1.616), <0.678 [0.220]

Trust level substantive doctor turnover rates 1.015 (1.009 to 1.021)*, <0.001 [0.003] 1.001 (0.998 to 1.004), <0.618 [0.001] 0.996 (0.988 to 1.004), <0.316 [0.004]

Trust level vacancy rates [full-time equivalent (FTE)] 1.000 (0.999 to 1.001), <0.463 [0.001] 0.999 (0.999 to 1.001), <0.784 [0.001] 0.999 (0.998 to 1.002), <0.762 [0.001]

Region

(reference region is London) Reference group Reference group Reference group

South West 0.676 (0.432 to 1.058), <0.087 [0.154] 1.453 (1.112 to 1.898)*, <0.006 [0.198] 0.618 (0.316 to 1.493), <0.227 [0.246]

South East 0.762 (0.522 to 1.115), <0.163 [0.148] 1.334 (1.043 to 1.706)*, <0.021 [0.167] 0.484 (0.252 to 1.092), <0.047 [0.177]

Midlands 1.134 (0.785 to 1.639), <0.501 [0.212] 1.419 (1.126 to 1.787)**, <0.003 [0.167] 0.522 (0.276 to 1.086), <0.066 [0.185]

East of England 0.872 (0.575 to 1.324), <0.521 [0.186] 1.526 (1.169 to 1.992)*, <0.002 [0.208] 0.400 (0.182 to 0.890)*, <0.025 [0.164]

North West 1.167 (0.794 to 1.717), <0.431 [0.229] 1.318 (1.033 to 1.679)*, <0.026 [0.163] 0.722 (0.412 to 1.773), <0.370 [0.262]

North East and Yorkshire 0.803 (0.527 to 1.223), <0.308 [0.173] 1.441 (1.114 to 1.865)*, <0.005 [0.190] 0.584 (0.269 to 1.230), <0.174 [0.231]

constant 0.211 (0.125 to 0.358)*, <0.001 [0.056] 0.428* (0.283 to 0.671), <0.001 [0.096] 0.174 (0.038 to 0.357)*, <0.001 [0.080]

a Model A included data on 220 trusts (observation) while models B and C included data on 214 trusts with robust standard errors.

b Coefficients can be interpreted as proportionate changes, for example, trusts that were rated as inadequate and requiring improvement had 45% higher locum intensity than trusts that were rated as having good and outstanding services.

Results Page 9 line 3 – Could the authors specify the share of locum shifts in comparison to all the shifts worked by medical staff who are not locum? This could help contextualise the findings and help further understand the extent of locum shift use. For instance, do such shifts represent 5, 10 or 15% of shifts worked by medical staff overall?

Response: Thank you for your comment. The number of shifts worked by permanent medical staff is not available and this is the reason our measure of locum intensity adjusted for permanent doctor FTE instead. NHS Improvement only collect data on the number of shifts filled by locums. NHS Digital collects data on FTE and headcount for all permanent doctors working in NHS trusts but does not collect any information on locum doctors. Therefore, the two organisations report their data in different units (shifts vs FTE) and this did not allow us to express the share of locum shifts in comparison to all shifts worked by all other medical staff. However, using permanent doctor FTE we adjusted the estimates of locum use and expressed rates of use in comparison to the work done by permanent doctors. For example, in p.9 par.3 (locum intensity results) we found that on average 4.4% of medical staffing in 2019 (i.e. locum intensity=0.22) was provided by locums doctors across NHS trusts. This is also acknowledged in the limitations (p.16 last para) and we added additional information to explain this. The new statement reads, “Second, although NHS Improvement collects data on the number of locum shifts, it does not collect the shift duration, locum FTE or the number of shifts filled by permanent doctors which would allow a more straightforward comparison with permanent doctor FTE. Therefore, we had to assume that shift lengths for permanent and locum doctors were broadly equivalent in order to estimate the proportion of medical staffing provided by locum doctors. Should data on the number of shifts filled by permanent doctors or data on locum FTE become available, this

limitation could be addressed”.

Regional variation in locum use The descriptive statistics were nicely and concisely written. Well done. Again, why not use the data from the years 2020 and 2021?

Response: Here we aimed to provide descriptive statistics and explore regional variation for the main year of analyses (i.e. 2019). This was before any impacts of the COVID 19 pandemic, which we didn't want to focus on in all the analysis. Regional variation was broadly similar in all years, as shown in the updated figure 2 in the supplement, and we feel that the descriptive statistics table should focus solely on the year 2019, for space reasons and the readability of the manuscript. Following your suggestion, we now provide data on regional variation in locum use, agency shifts and unfilled shifts for the years 2020 and 2021. (Tables 2 and 3 in the supplement).

Results from regression analyses I realise there are many results to present, but Table 3 could benefit from being re-arranged to be more reader-friendly. Perhaps using squared brackets for the robust standard errors and stars for the significance level. Also, the authors should briefly add the interpretation of IRR (e.g. meaning when the IRR are below or above 0). Note “b” could perhaps be moved up the table in the 1st paragraph. As for the result interpretation, should it not be as follows: North West had on average 4.5% higher locum intensity than trust in London, as the IRR=1.045?

Response: Thank you for these helpful suggestions. We have added square brackets for standard errors, and added asterisks to denote significant results. We added “b” from the footnote in the main text p.12 par.1. and we also provide a statement about the interpretation of IRRs which now reads “The results are reported as incidence rate ratios (IRRs) for the coefficients of interest followed by P-values and standard errors in square brackets and 95% confidence intervals in brackets. IRRs are defined as the number of exposed events (e.g. number of locum shifts) divided by the number of unexposed events (offset – e.g. permanent doctor FTE) in each time period and are essentially a ratio of two incidence rates. An IRR with a value greater than 1 indicates that the incident rate is higher in an exposed group compared to an unexposed group and the opposite is true for an IRR value less than 1”. We also corrected the typo in the footnote which now reads “Coefficients can be interpreted as proportionate changes, for example, trusts in the North West had on average 4.5% higher locum intensity than trusts in London.” We believe these changes have improved substantially table 2.

Table 2 – Locum intensity Can the authors explain why the CIs of ambulance trust services are so wide (IRR=55.43; 95% CI 20.56 to 149).? I'm sceptical as to whether such results are reliable. It may be of interest to present the confidence in squared brackets in the result section (e.g. 0.496 [0.299 – 0.258 95% CI]) to make it more reader-friendly.

Response: There were only 9 ambulance trusts in the data with high variability between them in terms of permanent doctor FTE and locum shifts. Ambulance trusts employed very few doctors (permanent doctor FTE mean=1.77 FTE [sd=1.40; median=1.29; 25th-75th percentile: 0.73 – 3.34]) and had few locum shifts (mean locum shifts=12.1 [sd=26.5; median=0; 25th – 75th percentile: 0 – 3.95]) in 2019. In p.16 par.1 we do acknowledge that these findings are a statistical artefact due to the very low number of permanent doctors and the very low number of locum shifts in these trusts. We amended the previous statement to make this more eminent in the manuscript. The statement now reads “However, this result is an artefact of the very low numbers of permanent doctors employed by ambulance trusts and the very small number of locum shifts filled when compared to other trust types.”

Table 2 – Agency shift. It would be worth mentioning that the results for the trust size are not significant.

Response: We have added a statement at the end of p.15 par.1, which reads, “The effects of trust size on the proportion of agency shifts were not statistically significant”.

Table 2 – Proportion of unfilled shifts. The result interpretations for NHS specialist trusts seem

incorrect. Should it not read that such trusts have a lower proportion of unfilled shifts IRR= 0.233 [0.091 to 0.598 95% CI] in comparison to NHS general acute trust?

Response: We apologise for this typo. We meant that NHS specialist trusts had 76.7% lower proportion of unfilled shifts when compared to NHS general acute trusts. This has been corrected in the manuscript.

Summary

These findings “Our findings show that on average 4.4% of medical staffing in NHS trusts in 2019 was provided by locum medical staffing” only appears in the summary and should be added in the results section as well. Can the authors elaborate when they state “. [...] and can provide important information about the effective planning of the NHS workforce”? What do the authors mean in terms of “effectiveness” (e.g. effective in terms of staff-per-patient ratio, saving costs or productivity?) As these aspects are not addressed in their paper I would suggest rephrasing by being more specific.

Response: Thank you, this is a very good suggestion. We have added a statement in the results section, p.8 par1. where we provide the average figure of locum intensity and explain how we calculated it. The statement reads “Assuming five shifts per permanent doctor FTE, the average trust level locum intensity of 0.22 locum shifts per permanent doctor FTE in 2019 is equivalent to 4.4% (i.e. $[0.22/5]*100$) of medical staffing provided by locums (25th–75th centile=2.2% to 6%) in that year.” We have elaborated on the statement about the effective planning of the NHS workforce. The new statement – p.15 par.3 - now reads, “Our findings can help inform NHS organisations about the extent of their locum use and provide for the first time important information about the drivers of locum use across NHS trusts. This can help with the effective planning of the NHS workforce by providing a better understanding of the make-up and spread of the locum medical workforce in England to aid recruitment in underperforming areas”.

Could the authors estimate (if data or available research permits) the extra financial costs of using agency locum to fill shifts, based on their regression results? For instance, trusts with worse CQC ratings use more locums: what would this represent in terms of pound sterling for that trust (e.g. extra £ pound spent)

Response: Due to the NHS Improvement database being relatively new, cost information for bank or agency locums is not available and we therefore cannot estimate the extra financial costs of using agency locum to fill shifts. We could provide a crude estimate of the cost but given the large fluctuations in locum costs, our estimate would not be accurate. We also added a statement in p.16 par. 2 - strengths and limitations – about the non-availability of cost data. This statement reads, “Fifth, the data do not contain any information on costs for locum doctors and we were therefore unable to estimate the extra financial costs of using locums to fill shifts.”

Strength and limitations - Line 34: I would suggest rephrasing the following sentence “We reveal the impact of COVID-19 on locum use in NHS Trust.” Considering you provided only descriptive statistics for the year 2020, your findings are not measuring the impact, per se. Rather you’re providing evidence on the extent to which locum staff were used during the pandemic.

Response: This is very good point. We deleted the previous statement and added a new statement which reads “We provide evidence on the extent of locum use across NHS Trusts during the COVID-19 pandemic.”

I would suggest using the same font for the last sections, contributors, ethics approval etc.

Response: Thank you. We have now applied the same font and font size to the last sections of the manuscript.

Figure 2 – whilst the legend states there is a red line, there are none on the graph. You should perhaps call it the dotted line. For each graph, could you add the names of the first, last and median trust names?

Response: We assume the reviewer is referring to figure 1, which depicts the three outcomes at the NHS Trust level in 2019. There was a typo in this figure referring to the line of the median for each outcome. We have amended the footnote in figure 1 from “red line” to “dash line” to correct this. Regarding adding the names of the first, last and median trusts to the figures, we would like to refer the reviewer to table \$1 in the supplement where we provide information on trusts with the highest and lowest locum intensity in 2019. We feel adding the trust names to figure 1 would distract from the message we are trying to convey with this figure, i.e. we would like the read to focus on the general pattern shown, and not the first/last trust.

Figures 1, \$1 and \$2 – Since you are comparing the years 2021 and 2021, would it be worth putting the graphs showing the same information next to each other for ease of comparison

Response: We considered arranging graphs in different order to enable readers to compare the outcomes across different years; however, the journal only allows a certain number of figures and tables in the main text. We attach two new figures where figures 1, \$1, \$2 and figures 2, \$3, \$4 are combined in a multi-graph format in the supplement.

Reviewer: 2

This is an interesting and well-written paper that examines, for the first time, the degree to which locums doctors in NHS trusts are used. The methods are clear, the results are clearly presented, and the discussion makes sense. Limitations are articulated. I have a few suggestions that may help improve clarity further.

1. It's a bit difficult to understand the logic of the several measures without first seeing the box. The authors might introduce the outcome measures section with the statement “a worked example of the algorithm...”, then go into the description of the measures.

Response: We have added the statement “We present a worked example of the algorithm that we used in the calculation of each outcome in Box 1 at the end of the first paragraph in p.7 (i.e. “Locum intensity”).

2. Within the box, there is a typo (page 17, line 11). I believe it should be ‘doctor FTE reported in that month.’

Response: Thank you for this. We have corrected the typo.

3. I'm having trouble reconciling the statement in the abstract that ‘over time, 2/3 of locum shifts...’ with the results section, page 8, line 4 ‘45.3% were bank shifts and 54.7% were agency.’ I see lower in that page (line 36) the 2/3 reference. Unless I'm misreading, those should all be consistent: they all represent the proportion of locum use that is agency vs. bank. Perhaps the distinction is in ‘total UNADJUSTED’ on line 3, page 8. If so, the authors should clarify in the abstract and the other sections that the 2/3 is an adjusted number.

Response: Thank you for bringing this to our attention, it is a very good point. The two figures refer to different things. In p.8. line 4 the two figures represent the total number of shifts for both bank and agency staff and are aggregated across all trusts in 2019. In the results section, ‘proportion of agency shifts’ the figures reported represent the mean proportion of agency shifts that each trust filled and that was calculated as the (number of agency shifts/number of bank staff shifts + number of agency shifts)*100. The latter figure captures the mean proportion of shifts that were filled by agency staff in 2019. We have added a statement in the abstract to clarify this where it now reads “Over time, on average two thirds of locum shifts were filled by locum agencies and a third by trusts’ staff banks”. We hope this clarifies the reviewer’s question.

4. Page 7, line 44 in the parentheses, I believe the authors mean that ‘8.6% of data were missing.’

Response: This typo has been corrected.

5. While the authors report IRRs that have very large percentage differences, it would be helpful to also provide an estimate of the absolute differences in shift coverages, which may not be quite as dramatic.

Response: Thank you, this is a very good suggestion. We estimated the absolute differences in shift coverages for the statistically significant coefficients across all models and the results are provided below. To do this we used the margins command in Stata, which estimates absolute differences in shift coverages for all three outcomes adjusted for all predictors in the models for the strata of interest. The table with the estimates of absolute differences in shift coverages is now provided in the supplement (table S5) and some results are discussed in the relevant sections of the manuscript.

Total shifts Agency shifts Unfilled shifts
CQC rates

Good + outstanding 170.5 (130 to 211.1), <0.001 [20.7] - -

Requiring improvement + inadequate 255 (197.1 to 312.9), <0.001 [29.6] - -

Trust type

NHS non-specialist 213.8 (168.9 to 258.6), <0.001 [22.9] 94.1 (86 to 102.3),

<0.001 [4.17] 31 (22.6 to 39.4),

<0.001 [4.31]

NHS specialist trust 61 (31.5 to 90.6), <0.001 [15.1] 142.2 (98.3 to 186),

<0.001 [22.4] 7.23 (0.53 to 14),

<0.001 [3.42]

Mental health trust 206.5 (122.7 to 290.3), <0.001 [42.8] 148.3 (112.4 to 184.3),

<0.001 [18.3] -

Ambulance service 11,829.7 (1,372.9 to 22,286.5), <0.001 [5,335.2] 3.2 (-1.51 to 7.82),

<0.185 [2.4] -

Community service 308.4 (127.5 to 489.3), <0.001 [92.3] - -

Trust size

Quintile 1 452.9 (201.3 to 704.6), <0.001 [128.3] - -

Quintile 2 355.3 (196.8 to 513.8), <0.001 [80.9] - -

Quintile 3 224.8 (150.9 to 298.7), <0.001 [37.7] - -

Quintile 4 276.9 (193 to 360.8), <0.001 [42.8] - -

Quintile 5 157.3 (111.2 to 203.5), <0.001 [23.6] - -

a Model A included data on 220 trusts (observation) while models B and C included data on 214 trusts with robust standard errors.

b Coefficients can be interpreted as absolute changes, for example, trusts that were rated as inadequate and requiring improvement on average had 84.5 more weekly locum shifts than trusts that were rated as having good and outstanding services.

References

1. Moynihan R, Sanders S, Michaleff ZA, Scott AM, Clark J, To EJ, et al. Impact of COVID-19 pandemic on utilisation of healthcare services: a systematic review. 2021;11(3):e045343.
2. Howarth A, Munro M, Theodorou A, Mills PR. Trends in healthcare utilisation during COVID-19: a longitudinal study from the UK. 2021;11(7):e048151.
3. <http://www.pulsetoday.co.uk/news/gp-partners-to-continue-offering-less-locum-work/20041228.article>.
4. The Nuffield Trust. The NHS workforce in numbers. The Nuffield Trust. 2022.
5. NHS Digital. NHS Workforce Statistics 2022. NHS Digital. 2022.

VERSION 2 – REVIEW

REVIEWER	Ejebu, Ourega-Zoé University of Southampton, School of Health Sciences
REVIEW RETURNED	16-Jan-2023
GENERAL COMMENTS	Dear authors, thanks for addressing my comments. The paper read very well. There is only a minor typo on page 70 line 45: [0.187 to 0.644). Please use bracket or parenthesis